# Role of Mucosal Protrusion Angle in Discriminating between True and False Masses of the Small Bowel on Video Capsule Endoscopy

**DOI:** 10.3390/jcm8040418

**Published:** 2019-03-27

**Authors:** May Min, Michael G. Noujaim, Jonathan Green, Christopher R. Schlieve, Aditya Vaze, Mitchell A. Cahan, David R. Cave

**Affiliations:** 1Department of Internal Medicine, University of Massachusetts Medical School, 55 Lake Ave N., Worcester, MA 01655, USA; 2Department of Internal Medicine, Duke University School of Medicine, 2301 Erwin Rd, Durham, NC 27705, USA; mgn9@duke.edu; 3Department of Surgery, University of Massachusetts Medical School, 55 Lake Ave N., Worcester, MA 01655, USA; jonathan.green@umassmemorial.org (J.G.); christopher.schlieve@umassmemorial.org (C.R.S.); mitchell.cahan@umassmemorial.org (M.A.C.); 4Division of Cardiology, University of California Irvine, 333 City Blvd W., Suite 400 Orange, CA 92868, USA; vazea@uci.edu; 5Division of Gastroenterology, University of Massachusetts Medical School, 55 Lake Ave N., Worcester, MA 01655, USA; david.cave@umassmemorial.org

**Keywords:** small-bowel mass, small-bowel bulge, video capsule endoscopy

## Abstract

The diagnosis of small-bowel tumors is challenging due to their low incidence, nonspecific presentation, and limitations of traditional endoscopic techniques. In our study, we examined the utility of the mucosal protrusion angle in differentiating between true submucosal masses and bulges of the small bowel on video capsule endoscopy. We retrospectively reviewed video capsule endoscopies of 34 patients who had suspected small-bowel lesions between 2002 and 2017. Mucosal protrusion angles were defined as the angle between the small-bowel protruding lesion and surrounding mucosa and were measured using a protractor placed on a computer screen. We found that 25 patients were found to have true submucosal masses based on pathology and 9 patients had innocent bulges due to extrinsic compression. True submucosal masses had an average measured protrusion angle of 45.7 degrees ± 20.8 whereas innocent bulges had an average protrusion angle of 108.6 degrees ± 16.3 (*p* < 0.0001; unpaired *t*-test). Acute angle of protrusion accurately discriminated between true submucosal masses and extrinsic compression bulges on Fisher’s exact test (*p* = 0.0001). Our findings suggest that mucosal protrusion angle is a simple and useful tool for differentiating between true masses and innocent bulges of the small bowel.

## 1. Introduction

The diagnosis of small-bowel tumors is challenging due to their low incidence, nonspecific clinical presentation, and the limitations of traditional endoscopic techniques. Video capsule endoscopy (VCE) has dramatically improved our ability to detect small-bowel tumors by enabling the visualization of portions of the small-bowel that are not accessible by colonoscopy or upper endoscopy [1]. VCE was able to diagnose small-bowel tumors in 8.9% of the 562 patients in a single-center retrospective study who underwent VCE for occult gastrointestinal bleeding, abdominal pain, and a variety of other indications [2]. Furthermore, VCE missed only 10% of small-bowel tumors compared to a collective miss rate of 73% by double balloon enteroscopy, small-bowel series, colonoscopy, and ileoscopy [3]. 

One of the major limitations of VCE is its inability to biopsy lesions identified during passage through the small bowel. Though double balloon enteroscopy can potentially be used to visualize the entire small intestine, reported rates for total enteroscopy are widely variable (ranging between 20 and 90%) and are highly user-dependent [1]. A group of experts at the 2006 International Conference on Capsule Endoscopy identified several major and minor characteristics of small-bowel lesions that are predictive of tumors, including mucosal disruption, bleeding, irregular surface, polypoid appearance, color, delayed passage, white villi, and invagination [4]. However, in the absence of these features, it can be challenging to differentiate between true submucosal masses and benign bulges arising from extrinsic compression by adjacent structures. 

In order to address this challenge, Girelli et al. developed the “smooth, protruding lesions index at capsule endoscopy” (SPICE) and examined its utility through a single-center, prospective study of 25 patients [5]. SPICE score was calculated by adding one point for each of the following: (1) Well-defined boundary with surrounding mucosa, (2) diameter less than height, (3) visible lumen, and (4) image of lesion lasting more than 10 min. A SPICE score >2 was found to be 83.3% sensitive and 89.4% specific for identifying true submucosal masses, therefore supporting a novel system for differentiating true from false masses on VCE. Through our retrospective study, we will evaluate the utility of an additional morphologic criterion, the mucosal protrusion angle. We have defined this as the angle between the small-bowel protruding lesion and surrounding mucosa. We hypothesize that false masses arising from extrinsic compression will create more obtuse protrusion angles >90° compared with true submucosal masses, <90°. By determining the utility of the mucosal protrusion angle, we hope to increase the specificity and sensitivity of VCE for detecting submucosal masses of the small bowel.

## 2. Experimental Section

### 2.1. Study Design

Patient demographics, indication for VCE, findings on VCE, radiographic studies, endoscopic and surgical interventions, pathology results, and survival following VCE were all collected retrospectively. Only those patients who were found to have a small-bowel protruding lesion on VCE were included in the study. Small-bowel protruding lesions were defined as any masses seen on VCE, including suspected submucosal masses and benign bulges. In total, we analyzed the VCEs of 34 patients. All VCEs were performed with the M2 A, PillCam^TM^ SB2 or SB3 (Medtronic, Minneapolis, MN, United States) and were analyzed using RAPID^TM^ version 8.3 (Given Imaging LTD, Yoqneam, Israel). This study was approved by the UMass Medical School Institutional Review board on December 2, 2015.

### 2.2. Angle Measurement

All angles were obtained through VCE images on RAPID^TM^ software version 8.3 (Given Imaging LTD, Yoqneam, Israel). The mucosal protrusion angle was defined as the angle between the protruding lesion and surrounding mucosa. Mucosal protrusion angles were measured using a protractor placed on the computer screen. We categorized lesions as having a protrusion angle of either >90° or <90° and hypothesized that an angle >90° suggests an external protrusion or bulge while an angle <90° suggests a submucosal mass. The frame for protrusion angle measurement was selected independently at each user’s discretion based on the frame in which they felt the protrusion angle could best be measured. A sample image with angle measurement technique was provided to each operator (see Figure 1). Angles were measured independently by two novice users and one expert user to assess for interobserver agreement. Both novice users performed <10 VCEs prior to this study and the expert user performed >1000 VCEs. 

### 2.3. SPICE Calculation

SPICE scores were calculated for each patient as outlined in Girelli et al. [5]. Lesions were given 1 point for the following: (1) Sharp boundary with surrounding mucosa, (2) height larger than diameter, (3) visible lumen in the frames in which the lesion appears, and (4) image of the lesion lasting more than 10 min. Any lesion with greater than two of the four SPICE criteria were predicted to be true submucosal masses per the findings in Girelli et al. A ruler placed directly on the computer screen was used to determine exact height and diameter of the small-bowel lesions.

### 2.4. Statistics

We calculated the sensitivity, specificity, positive predictive value (PPV), and negative predictive value (NPV) of both SPICE and protrusion angle. Fisher’s Exact Test was performed to assess the association between protrusion angle and true vs. false submucosal mass. All Fisher’s tests were one-tailed and the cutoff for significance was set at a p-value of <0.05. Interobserver agreement (kappa statistic) was assessed by comparing angle measurements of two novice VCE users and an expert user. We ran a logistic regression on capsule angle measurements for expert and novice users combined using a cutoff value of <90 degrees for a true mass and fit the data to a receiver operating characteristic (ROC) curve. We also ran a logistic regression on SPICE scores using a cutoff of >2 for true mass and fit the data to an ROC curve. Statistical analysis was performed using Stata Statistical Software: Release 13 (College Station, TX, USA).

## 3. Results

### 3.1. Demographics

We retrospectively reviewed the charts of 289 patients over the age of 18 who had undergone VCE for suspected small-bowel protruding lesions between January 2002 and March 2017. Of the patients, 241 were excluded because no protruding lesion was identified between the pylorus and ileocecal valve. Five patients were excluded because they were later identified as having true submucosal masses but did not have available pathology reports in our medical records. Nine patients were excluded because either the protrusion angle or SPICE score could not be determined due to poor image quality or limited visualization of the protruding lesion. In total, we analyzed the VCEs of 34 patients. The average age was 73.0 ± 16.6 years. There was a larger proportion of female patients (67.6%) compared with male patients (32.4%) (see Table 1)

### 3.2. Diagnosis

The most common indication for VCE was obscure gastrointestinal bleeding (41.2%), followed by abdominal pain (29.4.%). Twenty-five patients were found to have true submucosal masses based on pathology report (6 carcinoid, 2 gastrointestinal stromal tumor, 1 diffuse large B-cell lymphoma, 1 leiomyoma, 5 Peutz-Jeghers, 1 tubular adenoma, 1 hyperplastic polyp, 3 inflammatory polyps, 2 hamartomas, 1 lipoma, 1 cavernous hemangioma, 1 lymphangiectasia) and 9 patients had innocent bulges due to extrinsic compression (see Table 1). None of the patients with bulges had available pathology data because no mass was seen on follow-up studies such as enteroscopy or repeat capsule endoscopy.

### 3.3. Protrusion Angle and SPICE Calculations

True submucosal masses had an average measured angle of protrusion of 45.7° ± 20.80 whereas innocent bulges had an average protrusion angle of 108.6° ± 16.3° (*p* < 0.0001; unpaired *t*-test). When compared with SPICE scores, a mucosal protrusion angle <90° had a higher sensitivity (92.0% vs. 32.0%), PPV (96.0% vs. 88.9%), and NPV (66.7% vs. 32.0%). Both protrusion angle and SPICE scores had the same specificity of 88.9%. Acute angle of protrusion accurately discriminated between true submucosal masses and extrinsic compression bulges on Fisher’s exact test (*p* = 0.0001). Interobserver agreement between the two novice users and the expert user was good (κ = 0.67; 95% CI, 0.50–0.84). The area under the curve for mass angle using a cutoff value of <90 degrees for true mass was 0.93. The area under the curve for SPICE scores using a cutoff value of >2 for true mass was 0.55 (see Figure 2). 

## 4. Discussion

VCE has emerged as a convenient way to identify small-bowel tumors because it is non-invasive and allows for visualizuation of the entire length of the small bowel. Over the past several decades, its role in detecting malignancies has become more important as the incidence of small-bowel tumors has increased from 11.8 cases per million in 1973 to 22.7 cases per million in 2004. It is unclear how much of this increase can be attributed to improved diagnosis with the advent of VCE, however Bilimora et al. pointed to the rising incidence of carcinoid tumors as a major driving factor [6]. Prior studies have cited VCE malignant tumor detection rates as high as 63–83% [7,8]. In our study, we found a lower but still significant proportion of malignant tumors (45% of true submucosal masses) after excluding patients with Peutz-Jehgers. 

Though VCE has significantly improved our ability to detect small-bowel tumors, it has also opened up what Pennazio et al. describes as a “Pandora’s box” of findings including both malignant and benign lesions [9]. Bulges are among one of the most problematic benign findings on VCE, as they can often mimic the appearance of small-bowel tumors and contribute to false-positive outcomes [10]. False-positive outcomes may lead to further invasive and costly procedures, therefore highlighting the importance of differentiating bulges from true submucosal masses. Though “alarm” features, including bleeding, mucosal disruption, irregular surface, polypoid appearance, and white villi, have been described based upon expert consensus for malignant small-bowel masses, there are few studies available to support the use of these findings on VCE [4,11]. There have been prior attempts to use of automatic detection methods based on textural alterations on VCE, however none of these methods have been validated in clinical practice for diagnosing submucosal masses [12,13]. 

The SPICE score described by Girelli et al. was the first scoring system developed to distinguish between submucosal masses and bulges on VCE [5]. This study showed that a SPICE score >2 was highly sensitive (83.3%) and specific (89.4%) for detection of true submucosal masses. A validation study by Rodrigues et al. showed a lower sensitivity (66.7%) but high specificity (100.0%) for the SPICE score [14]. In our study, we found that a SPICE score >2 had an equally high specificity when compared with the mucosal protrusion angle but a significantly lower sensitivity of 32.0%. The discrepencies in our results may in large part be due to differences in study design, as the Girelli et al. study was prospective whereas ours was retrospective. Additionally, we included patients with Peutz-Jehgers and patients with “alarm” features outlined by Shyung et al., all of whom were excluded by Girelli et al. [5,11]. None of our true masses had a length time >10 min, criteria 4 on the SPICE scale, which made SPICE less sensitive in our patient population.

We evaluated the utililty of a new, simpler measure, the mucosal protrusion angle, in differentiating true masses from bulges. We found that an angle <90° accurately discriminated between true masses and extrinsic compression bulges (*p* = 0.0001). Acute protrusion angle also had a high sensitivity (92.0%) and specificity (88.9%) for distinguishing between true masses and bulges. It should be noted that we used both novice and expert users in our study, whereas the Girelli et al. study utilized only expert users. Discrepencies in angle measurements between the novice and expert users in our study were likely due to differences in the frames of the lesion on RAPID^TM^ chosen by each user. Despite these discrepencies, we found that there was good interobserver agreement between the novice and expert users when using mucosal protrusion angle (κ = 0.67; 95% CI, 0.50–0.84). This suggests that mucosal protrusion angle has the potential to be utilized by a wide range of users regardless of their VCE experience level.

## 5. Study Limitations

There are several limitations of this study that are important to note. First, this is a retrospective study and therefore is subject to both confounding and selection bias. As mentioned above, one potential source of bias is the variation in frame selection on VCE, as there was no way to ensure that all users would select the same image for angle measurement. In the future, it would be valuable to assess the degree of variability in frame selection between observers as this was not evaluated in our present study. An additional limitation of our study was that none of the bulges had pathologic confirmation due to our inability to visualize these transient lesions on subsequent interventions and the unethical nature of performing surgery or further invasive workup in such patients. We felt that long-term follow up provided an adequate surrogate but recognize this as a limitation. Finally, the number of patients in our study is comparatively small.

## 6. Conclusions

Mucosal protrusion angle is a novel and simple tool for differentiating between true masses and innocent bulges of the small bowel. To our knowledge, there are no prior studies examining the utility of this index. We found that small-bowel protruding lesions with a protrusion angle >90° are more likely to represent bulges and may not warrant any additional workup, whereas lesions with angle <90° are more likely to be true masses that should be evaluated for malignancy with enteroscopic or surgical interventions. Further prospective studies are still needed to validate our results.

## Figures and Tables

**Figure 1 jcm-08-00418-f001:**
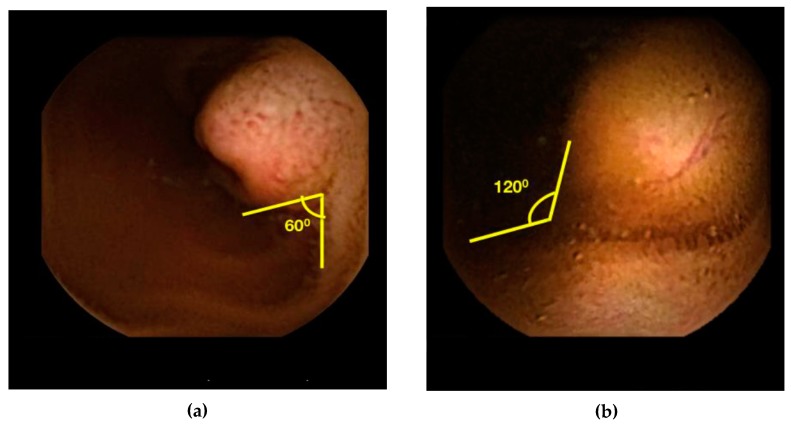
(**a**) Demonstration of acute angle measurement on RAPID^TM^. (**b**) Demonstration of obtuse angle measurement on RAPID^TM^.

**Figure 2 jcm-08-00418-f002:**
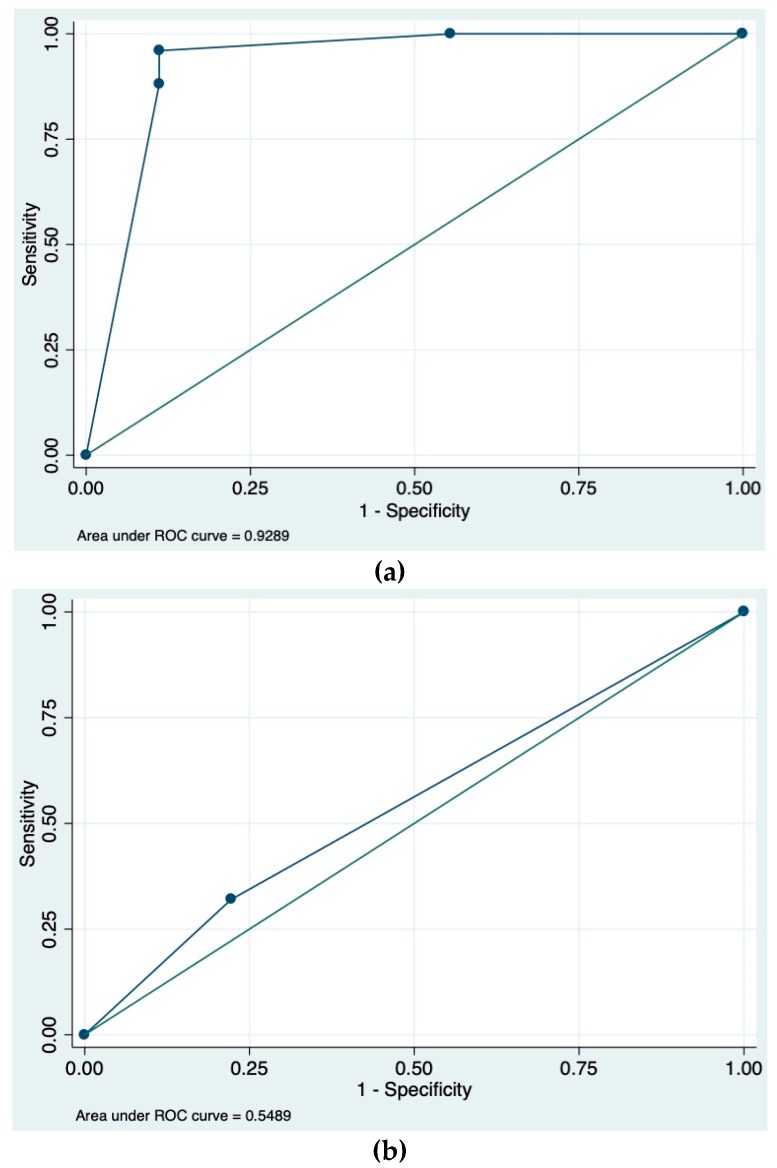
(**a**) The area under the receiver operating characteristic (ROC) curve for combined expert and novice mucosal protrusion angle using a cutoff of <90° for true mass. (**b**) The area under the ROC curve for smooth, protruding lesion at capsule endoscopy (SPICE) index using a cutoff of >2 for true mass.

**Table 1 jcm-08-00418-t001:** Patient Characteristics.

Gender	Age	Indication	Novice Angle ^b^	Expert Angle	Location	Imaging ^c^	Endoscopy ^c^	Surgery	Final Diagnosis
F	65	OGB	42.5	20.0	Jejunum	CTE +	ASBE +	Yes	GIST
M	52	OGB	16.0	10.0	Ileum	CT +	ASBE +	Yes	GIST
M	81	OGB	50.0	30.0	Ileum	CT −	ASBE −	Yes	Carcinoid
F	56	Carcinoid ^a^	20.0	10.0	Ileum	CT ±	Colo +	Yes	Carcinoid
F	77	IDA	27.5	20.0	Ileum	CT −	RSBE +	Yes	Carcinoid
F	56	CD	55.0	50.0	Ileum	CTE ±	RSBE +	Yes	Carcinoid
F	58	AP	10.0	20.0	Ileum	CT ±	Colo +	Yes	Carcinoid
F	62	AP	100.0	10.0	Ileum	CT +	Colo −	Yes	Carcinoid
M	38	AP	72.5	30.0	Jejunum	CT +	ASBE −	Yes	Inflammatory Polyp
F	73	OGB	35.0	110.0	Jejunum	ND	ASBE +	No	Lymphangiectasia
M	53	OGB	25.0	30.0	Ileum	CT −	Colo −	Yes	DLBCL
F	30	Peutz-Jeghers ^a^	45.0	30.0	Jejunum	ND	RSBE +	Yes	Peutz-Jeghers
F	39	Peutz-Jeghers ^a^	47.5	30.0	Ileum	ND	RSBE +	No	Peutz-Jeghers
F	36	Peutz-Jeghers ^a^	45.0	50.0	Duodenum	ND	ASBE +	Yes	Peutz-Jeghers
M	37	IDA	27.5	40.0	Jejunum	ND	ASBE +	Yes	Peutz-Jeghers
F	49	OGB	50.0	20.0	Jejunum	ND	ASBE +	No	Peutz-Jeghers
F	58	OGB	52.5	15.0	Jejunum	CT −	ASBE +	No	Inflammatory Polyp
M	37	Crohn’s ^a^	65.0	20.0	Jejunum	CT −	ASBE +	No	Inflammatory Polyp
F	76	OGB	40.0	40.0	Jejunum	CTE +	ASBE +	Yes	Hamartoma
F	57	OGB	45.0	10.0	Duodenum	ND	ASBE +	No	Hamartoma
M	78	BO	60.0	20.0	Ileum	MRE ±	ASBE −	Yes	Lipoma
F	83	AP	82.5	>90	Duodenum	ND	ASBE +	No	Tubular Adenoma
F	41	OGB	35.0	10.0	Ileum	ND	Colo −	Yes	Leiomyoma
F	48	OGB	47.5	10.0	Jejunum	ND	ASBE −	Yes	Hemangioma
M	47	AP	30.0	60.0	Duodenum	CT +	ASBE +	No	Hyperplastic Polyp
F	70	AP, OGB	130.0	70.0	Jejunum	CT −	ASBE −	No	Bulge
M	51	Leukemia ^a^	95.0	50.0	Jejunum	PET CT +	NA	No	Bulge
F	61	AP/CD	115.0	20.0	Duodenum	ND	ND	No	Bulge
F	29	AP	105.0	110.0	Ileum	CT −	Colo −	No	Bulge
F	55	OGB	75.0	20.0	Jejunum	NA	NA	No	Bulge
M	85	AP	122.5	130.0	Ileum	ND	Colo −	No	Bulge
M	29	AP	105.0	30.0	Ileum	CT −	Colo −	No	Bulge
F	54	OGB	125.0	130.0	Ileum	CT −	Colo −	No	Bulge
F	73	IDA	102.5	130.0	Ileum	CT −	ASBE −	No	Bulge

AP, abdominal pain; ASBE, anterograde small-bowel enteroscopy; BO, bowel obstruction; CD, chronic diarrhea; Colo, colonoscopy; CT, CT abdomen/pelvis; CTE, CT enterography; DLBCL, diffuse large B cell lymphoma; GIST, gastrointestinal stromal tumor; IDA, iron deficiency anemia; MRE, magnetic resonance enterography; NA, not available; ND, not done; OGB, obscure gastrointestinal bleeding; PET CT, positron emission tomography CT; RSBE, retrograde small-bowel enteroscopy. ^a^ Video capsule endoscopy performed for screening or surveillance. ^c^ Novice angle represents an average of measurement of 2 novice users. ^b^ Signs (+), (−), and (±) indicate positive, negative, and equivocal findings, respectively.

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
