# Peer review of "Role of Mucosal Protrusion Angle in Discriminating between True and False Masses of the Small Bowel on Video Capsule Endoscopy"

_jcm, 2019, doi:10.3390/jcm8040418_

Round 1
Reviewer 1 Report
This study shows the predictive value of a newly developed index, the mucosal protrusion angle, to identify true submucosal masses and to discriminate these from extramucosal bulging. The Actors show, by means of a retrospective analysis, that an open angle is associated to a extramucosal bulging, while closed angles indicate true submucosal lesions. This simple index, if validated prospectively in a a larger patients' series, could be useful in everyday clinical life and might reduce the cost of diagnoses and hospital stay. I have two major points to offer.
At VCE, mucosal lesions are often single frame images, or angles may appear differently according to intestinal motility and capsule movement. How did the Authors deal with these potential confounders? how may images were examined before choosing one? criteria must be given to ensure reproducibility.
I understand that included patients underwent surgery and histological confirmation was obtained. Yet, how many patients underwent other endoscopical (i.e. enteroscopy) or radiological ( TC, Entero-MR) examinations? was there an agreement between diagnostic procedures?
Minor: please check for typos
Author Response
Point 1: At VCE, mucosal lesions are often single frame images, or angles may appear differently according to intestinal motility and capsule movement. How did the Authors deal with these potential confounders? how may images were examined before choosing one? criteria must be given to ensure reproducibility.
Response 1: Operators independently chose the frame from which they felt the protrusion angle could best be measured. A sample image with angle measurement was provided (see Figure 1a and 1b), however there were no further instructions regarding frame selection. All the frames containing the lesions were examined by each operator, the number of images was highly variable from one image to several. This process has been further clarified in line 101-104 of the manuscript.
Point 2: I understand that included patients underwent surgery and histological confirmation was obtained. Yet, how many patients underwent other endoscopical (i.e. enteroscopy) or radiological ( TC, Entero-MR) examinations? was there an agreement between diagnostic procedures?
Response 2: All the patients had video capsule endoscopy, however most also underwent imaging and enteroscopic evaluation. We have added this information to Table 1.
Point 3: Minor: please check for typos
Response 3: We have reviewed the manuscript for typos prior to re-submission.
Reviewer 2 Report
The article is aimed to examine the utility of the mucosal protrusion angle in differentiating between true masses and bulges of the small-bowel on video capsule endoscopy. The title is “Role of Mucosal Protrusion Angle in Discriminating Between True and False Masses of the Small Bowel on Video Capsule Endoscopy”.
The study is a retrospective study.
A sample size of the study is relatively small.
Several factors influence the outcome of the study. Please discuss these factors.
Please review the literature and add more details in the discussion section.
Please add more details of the limitations.
What are the new knowledges from this study?
Finally, please recommend the readers “How to apply this knowledge for routine clinical practice?”.
Author Response
Point 1: Several factors influence the outcome of the study. Please discuss these factors.
Response 1: We agree with this statement and have expanded upon the Discussion section to further discuss the factors influencing the outcome of this study in line 231-236, 242-243, and 251-261.
Point 2: Please review the literature and add more details in the discussion section.
Response 2: We have reviewed additional studies in the literature and included them in the Discussion section in line 212-217.
Point 3: Please add more details of the limitations.
Response 3: We have added additional details about the study’s limitations in line 251-261.
Point 4: What are the new knowledges from this study?
Response 4: We have further highlighted the findings of the study in the Conclusions section from line 265-270. To our knowledge, there are no other studies looking at protrusion angle for differentiating between true submucosal masses and bulges. Therefore, we feel the finding that protrusion angle could be a useful index in video capsule endoscopy is novel.
Point 5: Finally, please recommend the readers “How to apply this knowledge for routine clinical practice?”.
Response 5: Based on our findings, we concluded that masses with a protrusion angle > 90 degrees are more likely to represent bulges and may not warrant any additional workup, whereas masses with angle < 90 degrees are more likely to be true masses that should be evaluated for malignancy with enteroscopic or surgical interventions (line 267-270).
Reviewer 3 Report
Authors present a study on the differential diagnosis of small bowel CE images suggestive of subepithelial lesion. the calculated the SPICE index and evaluated the mucosal protrusion angle.
Methods and results show contradictions & sometimes unclear:
1) Methods: "small bowel protruding lesions were included" - Aim: "differentiate between submucosal masses and benign bulges".
2) 12 pts had mucosal lesions (polyps) and should be excluded.
3) Method: "Patients with a suspected submucosal mass who did not have small bowel biopsies or surgical pathology were excluded from this study" - Results: " 9 patients had innocent bulges due to extrinsic compression (See Figure 3). None of the patients with bulges had available pathology data because no mass was seen on follow-up studies such as enteroscopy or repeat capsule endoscopy".
Some data needs to be included:
1) Inter- and intra-observer agreement on: 1) frame selection, 2) angle measurement; 3) between novice and expert.
2) ROC curve should be calculated for protrusion angle and SPICE index.
3) A table should be included with patient and lesion features, protrusion angle, DBE outcomes, interobserver and intraobserver agreement.
4) Bibliografy is incomplete.
Author Response
Point 1: Methods and results show contradictions & sometimes unclear:
Methods: "small bowel protruding lesions were included" - Aim: "differentiate between submucosal masses and benign bulges".
Response 1: We defined small-bowel protruding lesions as any masses seen on VCE, including both submucosal masses (“true” masses) and benign bulges (“false” masses). This was clarified in line 87-88. We also reviewed and edited these terminologies throughout the manuscript for consistency.
Point 2: 12 pts had mucosal lesions (polyps) and should be excluded.
Response 2: We did not exclude patients with polyps (i.e. hyperplastic polyp, hamartoma, lipoma, etc) because we considered these to be true submucosal masses. Though these lesions are not considered malignant, we felt they should still be differentiated from bulges.
Point 3: Method: "Patients with a suspected submucosal mass who did not have small bowel biopsies or surgical pathology were excluded from this study" - Results: " 9 patients had innocent bulges due to extrinsic compression (See Figure 3). None of the patients with bulges had available pathology data because no mass was seen on follow-up studies such as enteroscopy or repeat capsule endoscopy".
The first sentence in the Methods section was erroneous and deleted (line 88-89). We did include patients without pathology data, specifically in the bulge group.
Point 4: Some data needs to be included: Inter- and intra-observer agreement on: 1) frame selection, 2) angle measurement; 3) between novice and expert.
Response 4: We found there to be substantial interobserver agreement on angle measurement between novice and expert with kappa = 0.67 (95% CI 0.50-0.84). We included this finding in line 179-180. Unfortunately, we were not able to calculate interobserver agreement on frame selection as this data was not recorded in the original design of our study, this limitation has been added in line 252-254. We were also unable to calculate intraobserver agreement on frame selection and angle measurement as our users only measured the angles one time.
Point 5: ROC curve should be calculated for protrusion angle and SPICE index.
Response 5: We have included ROC curves for protrusion angle and SPICE index in figures 2a and 2b. We found the area under the ROC curve for protrusion angle to be 0.93.
Point 6: A table should be included with patient and lesion features, protrusion angle, DBE outcomes, interobserver and intraobserver agreement.
Response 6: We have added Table 1 which includes patient and lesion characteristics.
Point 7: Bibliografy is incomplete.
Response 7: The bibliography was reviewed for completeness prior to resubmission.
Round 2
Reviewer 2 Report
Overall, the revised version is O.K.